# Consideration of Intestinal Failure in Cases of De-Adaptation of Short Bowel Syndrome: A Case Report and Descriptive Review

**DOI:** 10.3390/healthcare9121660

**Published:** 2021-11-30

**Authors:** Tasuku Kato, Yasuhisa Nakano, Fumiko Yamane, Ryuichi Ohta, Chiaki Sano

**Affiliations:** 1Department of Medicine, Faculty of Medicine, Shimane University, 89-1 Enya cho, Izumo 693-8501, Japan; tasukno0619@gmail.com (T.K.); yasunakano11@gmail.com (Y.N.); 2Community Care, Unnan City Hospital, 699-1221 96-1 Iida, Daito-cho, Unnan 699-1221, Japan; fummy811@gmail.com; 3Department of Community Medicine Management, Faculty of Medicine, Shimane University, 89-1 Enya cho, Izumo 693-8501, Japan; sanochi@med.shimane-u.ac.jp

**Keywords:** intestinal failure, de-adaptation of short bowel syndrome, diarrhoea, ageing

## Abstract

Short bowel syndrome (SBS) causes malabsorption due to extensive intestinal resection. While intestinal function declines with age, little is known about the relationship between intestinal failure and ageing. For the first time in Japan, we report a case of de-adaptation of SBS thought to be due to ageing, in a 93-year-old woman who presented with electrolyte imbalance and malnutrition. She had undergone five surgical resections of the small intestine over the past 20 years. She had developed SBS once due to multiple surgeries, but due to compensatory function, the symptoms had abated. However, due to decreased intestinal function caused by ageing, it worsened and symptoms reappeared. A literature search for the period January 1990 to May 2021 in *Ichushi* a major journal in Japan, found that de-adaptation of SBS occurred in 23 previous cases, of which we were able to confirm the details in 17 cases, with no case reports on “de-adaptation of SBS”, demonstrating that the concept of “intestinal failure” has only recently begun to be used in routine practice. Therefore, we stress the importance of re-emphasizing the concept of ”intestinal failure” in everyday practice, as well as other organ-related conditions such as cardiac or renal failure, as this may lead to a better understanding of the pathogenesis of malnutrition and diarrhoea in elderly patients.

## 1. Introduction

The intestine is critical for the absorption of nutrients, such as sugars, proteins, electrolytes, and water. It is also thought to be an important organ for maintenance of systemic immune homeostasis; in fact, impairment of the gastrointestinal system has been shown to cause deterioration of immunological systems [1]. Short bowel syndrome (SBS) is a condition in which malabsorption occurs due to extensive intestinal resection. Typically, diarrhoea occurs after surgery, resulting in malnutrition, electrolyte imbalances, and vitamin deficiencies. Vitamin deficiencies that occur mainly involve fat soluble vitamins and can lead to problems such as osteoporosis and bleeding tendencies. Importantly, vitamin B12 deficiency that may occur in patients who have undergone terminal ileum resection can lead to macrocytic anaemia and neurological disorders [2,3].

Nevertheless, after excision, the intestine can compensate for intestinal function by increasing villus height, crypt depth, and the number of nutrient transporters, and accelerating intestinal cell differentiation [4,5]. However, extensive resection with an intestinal length of approximately 200 cm or less remaining cannot compensate for intestinal function, and results in SBS [2]. Moreover, comorbidities can modify gastrointestinal function too, further deteriorating the symptoms of SBS.

Intestinal function is known to decline with age. With ageing, changes in intestinal stem cells cause a reduction in the regenerative capacity and function of the intestine [6]. Furthermore, intestinal permeability increases, while absorption is reduced, due to the overproduction of proinflammatory cytokines such as IL-6, TNF-α, and IL-1β as a result of ageing. Finally, ageing also leads to changes in intestinal bacterial flora (increase in lipopolysaccharide-producing Gram-negative bacilli), intestinal inflammation, and a decline in intestinal function. However, not all the mechanisms have been elucidated [7].

SBS can lead to intestinal failure, in which the intestinal function is insufficient to meet the demands of digestion and absorption. The concept of “ntestinal failure” was first proposed by Fleming and Remington in 1981. However, this concept was rarely used until the European Society for Clinical Nutrition and Metabolism (ESPEN) established formal definitions and classifications. The condition “intestinal failure” is defined by ESPEN as: “the reduction of gut function below the minimum necessary for the absorption of macronutrients and/or water and electrolytes, such that intravenous supplementation is required to maintain health and/or growth” [8]. However, little is known about the relationship between intestinal failure and ageing. Nevertheless, based on the mechanisms mentioned above, it is plausible that ageing may lead to intestinal failure. 

Though the concept of “intestinal failure” is important when considering the nutritional status of the elderly, few studies have discussed ageing and SBS or intestinal failure. To that end, for the first time in Japan, we report a case of de-adaptation of SBS thought to be due to ageing and provide a review of the current literature on de-adaptation of SBS, to assess how many reports of de-adaptation of SBS with age have been published. Finally, we discuss the pathogenesis that the decline in intestinal function due to ageing causes intestinal failure. 

## 2. Case Presentation

### Patient

A 93-year-old patient visited our hospital with hypokalaemia, malnutrition, and decreased renal function detected by a family physician.

Five years before her visit to the hospital, she had undergone bowel resection several times (Table 1). As a result, she had been suffering from diarrhoea for about three months, thought to be caused by SBS. The diarrhoea improved spontaneously and she had no abdominal symptoms. Then, one year before admission, watery diarrhoea appeared, and although antidiarrhoeal medication was prescribed, there was little improvement.

Her past history included colonic perforation, abdominal wall hernia with strangulated ileus, and resection of about 2 m 30 cm (59.1 inches) of the terminal ileum (Figure 1). Five years prior to this admission, she was diagnosed with strangulated ileus, and the small intestine was resected, 7 cm from the terminal ileum and 50 cm from the ligament of Treitz (Figure 2). At presentation, the patient’s blood pressure was 95/67 mmHg, heart rate was 59 beats per minute, SpO_2_ as 95%, and her temperature was 36.6 °C. On physical examination, normal breath sounds and heart sounds with mild systolic murmurs were observed. The abdomen was flat and soft. Murphy’s sign was negative, and there was no costovertebral angle tenderness. Lower leg oedema was observed. The results of blood tests were as follows: white blood cell count 15.30 × 10^3^/μ (neutrophils 78.3%, lymphocytes 15.5%, monocytes 5.6%, eosinophils 0.4%, basophils 0.2%), red blood cell count 3.34 × 10^6^/μ, hemoglobin 11.3 g/dL, hematocrit 33.2%, platelet count 27.9 × 10^4^/μ, total bilirubin 1.6 mg/dL, aspartate aminotransferase (serum glutamic-oxaloacetic transaminase) 48 IU/L, alanine aminotransferase (serum glutamic-pyruvic transaminase) 37 IU/L, total protein 5.2 g/dL, albumin 2.5 g/dL, blood urea nitrogen 18.7 mg/dL, creatine 1.13 mg/dL, Na 143 mEq/L, K 1.7 mEq/L, Cl 99 mEq/L, Ca 5.0 mg/dL, P 2.8 mg/dL, Mg 0.8 mg/dL, and estimated glomerular filtration rate 34.1 mL/min/L. An abdominal computed tomography showed mild oedema of the colon, and no other obvious abnormalities (Figure 3). No obvious organic abnormalities were noted on imaging, but the bowel was shortened because of repeated bowel resections. Colonic specimens showed only nonspecific inflammatory findings (Figure 4a,b). We believe that this inflammatory finding was due to an imbalance between the protective and aggressive factors of the intestinal tract due to ageing.

After hospitalization, her electrolyte imbalance and renal function were improved by an infusion of magnesium and potassium, but the diarrhea persisted. A colonoscopy was performed and pathological specimens were taken. However, the colonoscopy, which was performed to evaluate diarrhea, did not reveal any specific findings. In addition, stool culture and histopathological examination did not reveal the exact cause. Taken together, this was likely to be de-adaptation of SBS, as a part of the clinical course. Since diarrhea and electrolyte abnormalities continued, we concluded that her bowel could not absorb nutrients, and therefore switched to central venous (CV) nutrition (delivered via a CV port), which improved her diarrhea. She was discharged in good general condition about one month after creation of the CV port.

## 3. Descriptive Review

A literature search for the period from January 1990 to May 2021 in *Ichushi*, a major journal in Japan, with the keywords: tanchoshyoukoku-gunn (“SBS” in English) AND saihatsu (“de-adaptation” in English), yielded 23 cases of de-adaptation of SBS (Figure 5). Of the 23 cases, we were only able to confirm the details in 17 cases, and no cases were identified that contained detailed information on recurrent cases of SBS (Table 2). 

## 4. Discussion

In this case, using the concept of “intestinal failure” to understand the pathological condition in the patient, an appropriate diagnosis and treatment were achieved. Importantly, we believe that the paucity of case reports on the de-adaptation SBS is not due the condition being rare. Rather, it is likely due to the fact that the concept of “intestinal failure” has only recently begun to be used in routine practice. In this case, the patient had SBS following previous bowel resections. Afterward, likely due to compensatory mechanisms occurring in the intestine, the patient’s SBS improved and she was symptom-free. However, the SBS recurred, and the patient went from a stable to an unstable state. The cause of this change is not yet clear. However, we believe that it is due to the effects of ageing. The following are some of the mechanisms that may have contributed to the decline in bowel function with age, and thereby worsened intestinal failure. As an individual ages, abnormalities in the intestinal epithelium occur due to age-related changes in stem cells, cytokines, and gut microbiota [6,7]. As a result, the absorption capacity is reduced and the immune barrier is incomplete, which can lead to pathological conditions, as was observed in this case. However, this study is not without limitations. First, the literature review in this study was limited to Japanese literature, and the number of reports was also limited. Therefore, it is necessary to clarify these details by increasing the number of cases studied and reported in the future.

## 5. Conclusions

This is the first report of de-adaptation of SBS in an elderly patient due to age-related loss of bowel function. Importantly, the introduction of the concept of “intestinal failure” into everyday practice as well as other organ-related conditions such as cardiac or renal failure may lead to a better understanding of the pathogenesis of malnutrition and diarrhoea in elderly patients. In the future, it is important to raise awareness regarding the pathogenesis of “intestinal insufficiency” and to train personnel and multidisciplinary teams capable of managing it appropriately. 

### Lessons Learned

To propose the concept of “intestinal failure”:

First, we stress the importance of raising awareness regarding the concept of “intestinal failure” and re-emphasizing it in everyday practice, as the introduction of this concept can lead to a better understanding of the patient’s condition and help inform treatment strategies. It can also allow for the early recognition and prevention of complications associated with intestinal failure. The main complications of intestinal failure include intestinal failure-associated liver disease, bone disease, and renal failure. Additionally, complications associated with long-term intravascular catheterisation can also occur. Importantly, intestinal failure-associated liver disease mainly occurs in children and can be caused by a variety of factors, including malnutrition, sepsis, and bacterial overgrowth. Bone disease is associated with nutrient deficiencies, whereas renal failure is associated with chronic dehydration [26,27]. Notably, these complications can be reduced with appropriate medical interventions. Therefore, the management of patients with intestinal failure needs to be carried out by a multidisciplinary team of experts. Nevertheless, the concept of intestinal failure is still not well understood by healthcare professionals in Japan. Therefore, a major future challenge will be to train personnel who can appropriately manage the condition.

2.Considerations for diarrhoeal symptoms in the elderly:

This case suggests that the pathogenesis of unexplained diarrhoea in the elderly, a scenario often encountered in daily practice, may be related to intestinal failure. Importantly, two major conditions are thought to be involved in diarrhoea in elderly individuals. The first is the effect of ageing. In fact, previous studies have shown that the integrity of the intestinal epithelial barrier and its regulatory function are reduced with ageing [1]. This change can lead to microbiota changes, increased intestinal permeability, and decreased absorptive function accompanying the overproduction of inflammatory cytokines such as IL-6, TNF-α, and IL-1β with ageing, thereby causing diarrhoea [7]. Second, the effect of invasion, such as surgery, as in this case, or sepsis, can lead to further deterioration. Notably, in the United States, the incidence of sepsis is roughly 535 cases per 100,000 person-years, and hospitalisations for sepsis exceed those for myocardial infarction and stroke [28]. Diarrhoea is often included as one of the symptoms of sepsis [29], and diarrhoea during sepsis is often thought to be caused by the use of antimicrobial agents [30]. However, as mentioned above, diarrhoea may also be caused by disruption of intestinal bacterial flora and increased intestinal permeability due to persistent inflammation. Therefore, the presentation of diarrhoea can be also be explained by the concept of intestinal failure. Thus, it is important to expand the differential diagnosis of diarrhoea to include intestinal failure. In fact, if the patient’s diarrhoea is caused by intestinal failure, the symptoms may persist even after the acute disease is cured. Additionally, especially in the elderly, diseases such as sepsis, in addition to ageing, may accelerate the decline in intestinal function and worsen the condition of intestinal failure. Considering the rapidly ageing population worldwide, it is important to take the concept of “intestinal failure” into consideration when investigating the causes of older patients’ diarrhoea.

## Figures and Tables

**Figure 1 healthcare-09-01660-f001:**
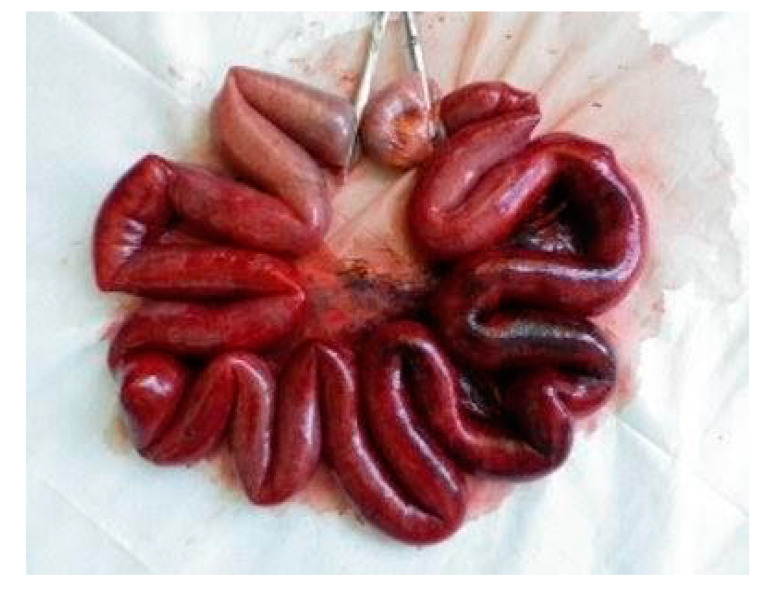
A small intestine of about 2 m 30 cm was removed from the terminal part of the ileum.

**Figure 2 healthcare-09-01660-f002:**
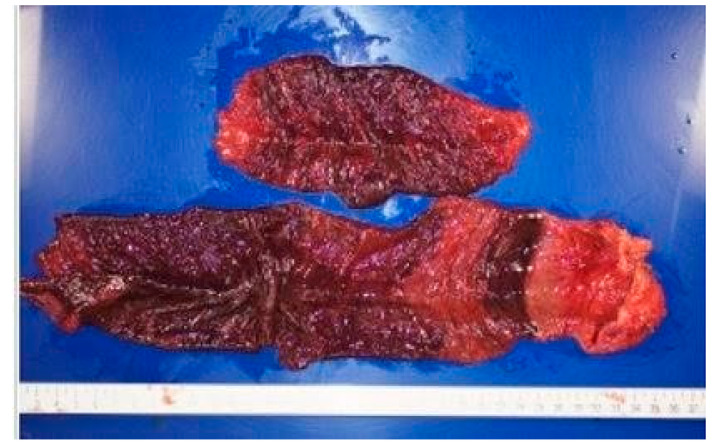
The small intestine was resected as follows: 7 cm from the terminal ileum and 50 cm from the Treitz ligament.

**Figure 3 healthcare-09-01660-f003:**
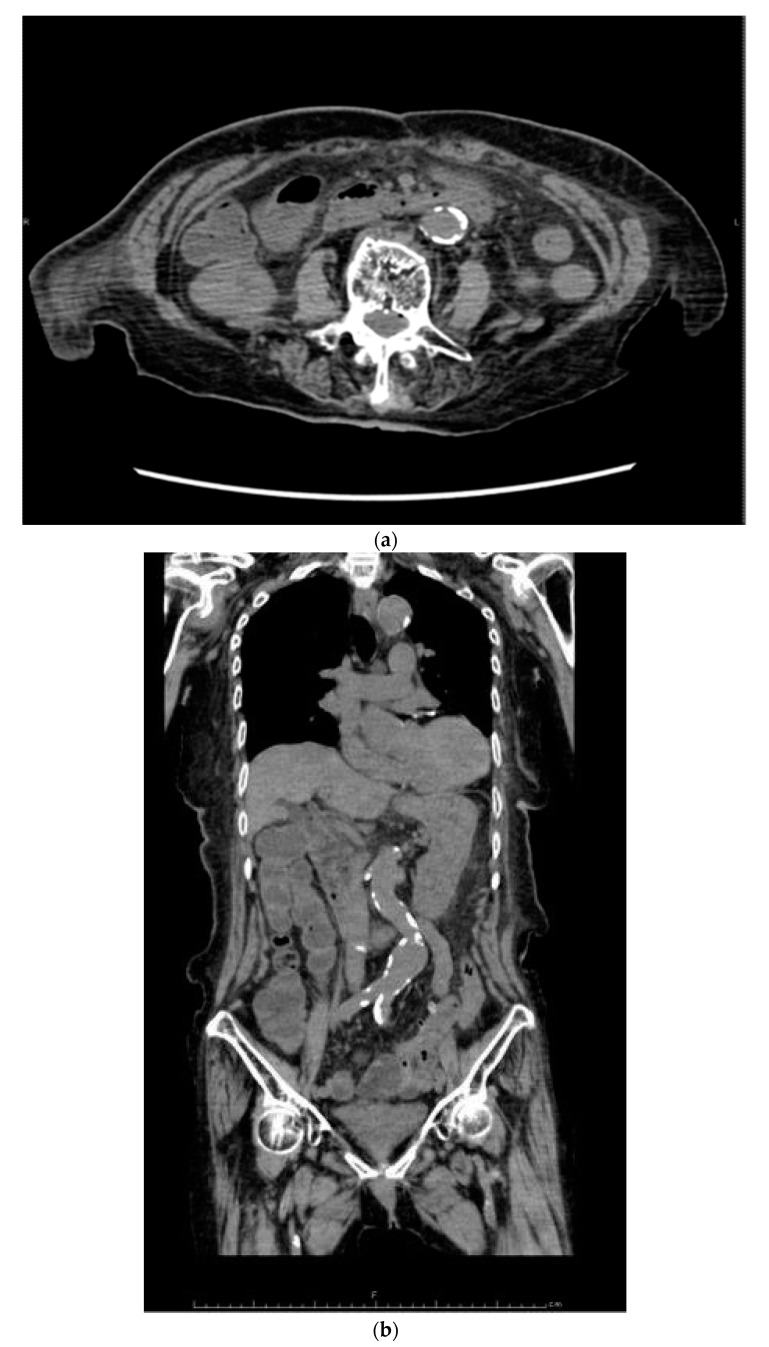
Findings from the abdominal computed tomography scan. Most of small intestine lacked in (**a**,**b**).

**Figure 4 healthcare-09-01660-f004:**
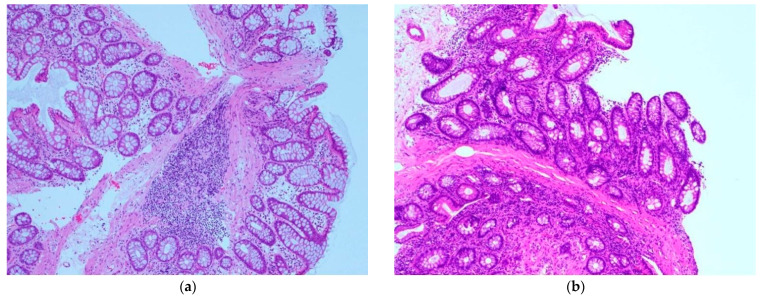
Colonic specimens. From (**a**,**b**), Infiltration of Inflammatory cells, lymphocytes and plasma cells, partly eosinophils was seen in mucous tissue. There is no obvious malignant findings.

**Figure 5 healthcare-09-01660-f005:**
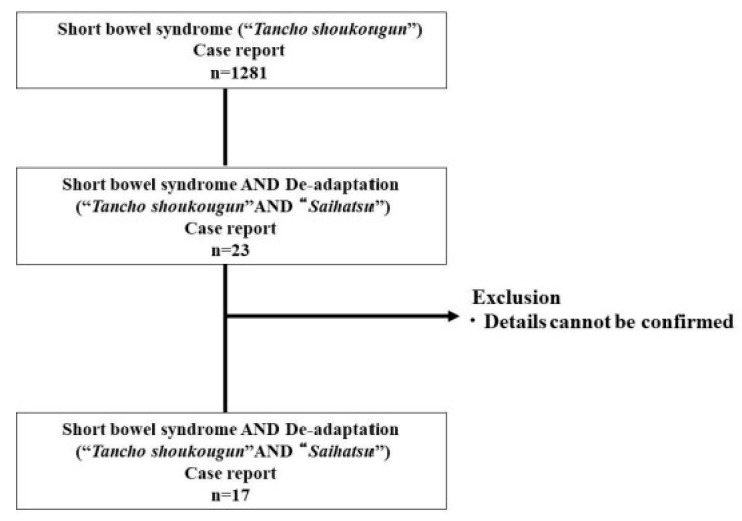
Search flow for case reports of “de-adaptation of SBS”.

**Table 1 healthcare-09-01660-t001:** Past history.

The Time of the Event (Based on the Time of Visit to Our Hospital)	Cause	About the Resected Bowel
20 years ago	sigmoid colon perforation	Details unknown
15 years ago	abdominal incisional hernia	Details unknown
10 years ago	abdominal incisional hernia	Details unknown
9 years ago	abdominal incisional hernia	Details unknown
7 years ago	abdominal incisional hernia (strangulated ileus )	About 2 m 30 cm of the small intestine was excised.(Figure 1)
5 years ago	abdominal incisional hernia (strangulated ileus)	The small intestine was excised as follows: 7 cm length from the terminal ileum and 50 cm from the ligament of Treitz. (Figure 2)

**Table 2 healthcare-09-01660-t002:** Available literature on the recurrence of short bowel syndrome in *Ichushi*.

Case No.	*Ichushi* No.	Reference	Age	Gender	Type of Medical Treatment	Length of Resected or Remaining Bowel	Survival Status
1	2020152323	[9]	70s	M	Suspected CV port infection, started cefmetazole.	Not mentioned.	The patient was discharged without fever.
2	2019013729	[10]	70	F	Diagnosed as superior mesenteric artery thrombosis, warfarin dose was increased step by step, and PT-INR was controlled at around 2.0 with 10 mg/day.	The only remaining small intestine was the jejunum 20 cm below the ligamentum flavum.	No bleeding tendency was observed, and the patient was discharged home after HPN management was introduced.
3	2017233400	[11]	75	F	Prophylactic ELT in an adult patient with short bowel syndrome undergoing home CV nutrition with repeated CRBSI.	Approximately 50 cm, from the Treitz ligament to the middle of the transverse colon, was resected.	No CRBSI or adverse events occurred during the first 12 months after the introduction of ELT.
4	2016233249	[12]	62	M	An oral solution of Se (50 μg/mL) and an injectable solution of Se (50 μg/mL) were prepared and administered to patients with refractory intestinal fistula and short bowel after recurrent rectal cancer surgery.	Not mentioned.	After 4 months, the Se blood level increased to 8.2 μg/dL, tachycardia symptoms disappeared, and the patient was able to walk unaided.
5	2015051943	[13]	62	F	Iron was discontinued for high FGF23 levels.	Not mentioned.	Electrolyte abnormalities were corrected and the patient was discharged after mild improvement.
6	2014157682	[14]	70	M	The visiting nurse station that had been visiting the patient prior to hospitalization had concerns about the management of the patient. Therefore, we requested another home nursing station with a certified nurse to intervene.	Not mentioned.	A smooth transition to home was achieved.
7	2014157563	[15]	68	M	The patient underwent massive resection of the small intestine, partial colonic resection, and jejunostomy. About 3 months later, the patient underwent a jejunojejunostomy and gastrostomy, and systemic management for postoperative short bowel syndrome was performed.	Only about 30 cm of the Treitz ligament remained.	The patient was doing well five years after the surgery.
8	2013302268	[16]	83	M	The patient was started on oral polaprezinc and intravenous infusion of trace elements in the high-calorie infusion.	Resection of the transverse colon from the small intestine approximately 20 cm from the Treitz ligament.	About one month later, the zinc and copper levels normalized, the anemia improved, and the white blood cell and neutrophil counts recovered to the normal range. No recurrence was observed afterwards.
9	2012211980	[17]	43	F	The patient underwent emergency extensive small bowel resection, and the small bowel segment was closed without anastomosis. After confirming the disappearance of the leftover thrombus, a small bowel anastomosis was performed. The patient was then treated with oral intake and intermittent central venous nutrition.	SBS with 10 cm of residual small bowel.	Three years after surgery, the patient was still alive without recurrence.
10	2012155105	[18]	78	M	The patient was diagnosed with small bowel perforation due to recurrent NOMI. A repeat laparotomy was performed.	SBS with residual small intestine of approximately 50 cm.	Enteral feeding was started after complete recovery.
11	2011219727	[19]	49	F	After 14 months of chemotherapy for anaplastic pleomorphic sarcoma, the patient developed peritoneal recurrence.	Remaining jejunum was 70 cm long.	She developed symptoms of intestinal obstruction, and after systemic management with home CV nutrition, her general condition deteriorated and she expired.
12	2011155844	[20]	21	M	The patient had been complaining of chronic abdominal pain and bloating. As a preventive measure, we administered probiotics (*Bifidobacterium breve Yakult strain (BBG-01)*), which markedly improved her abdominal symptoms and intestinal microflora.	Remaining small intestine was about 30 cm.	Four years later, the abdominal symptoms had stabilized and the intestinal pneumoperitoneum had not recurred.
13	2010270455	[21]	49	F	Because diverticulitis and necrotic perforation of the gastrointestinal tract were observed in the ileum, a partial small bowel resection was performed.	Intestinal resection of 40 cm of oral ileum from the ileocoecal valve. Ten months later, the patient underwent an ileal resection of 20 cm from the ileocoecal valve due to diverticulitis.	The postoperative course was favourable, and the patient was placed on outpatient follow-up.
14	2009043943	[22]	1 year and 3 months	M	At 5 days of age, a boy underwent Ladd’s operation for midgut axis torsion due to abnormal bowel rotation.	Remaining small intestine was about 40 cm.	Intestinal ischemia did not improve after removal of the torsion, and the necrotic intestine was resected. The remaining intestine was 40 cm in length, and the patient suffered from short bowel syndrome after the operation.
15	2008315742	[23]	56	M	The patient was diagnosed as having generalized peritonitis due to tumour perforation and underwent emergency mesenteric tumour resection, partial jejunal resection, intraperitoneal lavage, and drainage.	At the time of initial surgery: The mass was 8 × 8 × 5 cm in size and weighed 820 g. At reoperation: the small intestine was resected from the Treitz ligament on the oral side and from the end of the ileum to 35 cm on the anal side. The small intestine was resected from the Treitz ligament on the oral side and from the end of the ileum to 35 cm on the anal side.	The patient died 5 months after reoperation due to deterioration of nutritional status caused by short bowel syndrome.
16	2005259857	[24]	51	F	Although massive bleeding occurred, the remaining small intestine was approximately 100 cm long after complete removal of the tumour.	Remaining small intestine was about 100 cm.	Four months later, dietary intake and diarrhoea were well controlled, and 20 months passed since the surgery.
17	2005214149	[25]	59	M	The patient underwent only a partial resection of the small intestine near the ileal transition of the jejunum, where the large tumuor was concentrated.	Partial resection of approximately 30 cm of the small intestine in the vicinity of the jejunal–ileal transition. Partial resection of the small intestine.	Postoperatively, the patient was treated with imatinib mesylate, and no recurrence was observed

M, male; F, female; SBS, short bowel syndrome; CV, central venous; HPN, home parenteral nutrition; ELT, endoscopic laser therapy; CRBSI, catheter-related bloodstream infection. NOMI, non-occlusive mesenteric ischemia.

## Data Availability

The datasets used and/or analysed during the current study may be obtained from the corresponding author upon reasonable request.

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
