# Peer review of "Consideration of Intestinal Failure in Cases of De-Adaptation of Short Bowel Syndrome: A Case Report and Descriptive Review"

_healthcare, 2021, doi:10.3390/healthcare9121660_

Round 1
Reviewer 1 Report
Dear authors
First, I would like to congratulate you for highlighting a new concept of recurrence of short bowel syndrome. Also, linking it to aging is another new concept. However, I have various comments that must be addressed.
Major issues
- First, Intestinal failure is not a new concept. In fact, it is quite commonly used in children. Please avoid using the words 'introducing the concept of Intestinal failure'. Rather, you are re-emphasizing it.
- Second, the recurrence of SBS is again a very strong statement. I think you should mention it as "de-adaptation". Adaptation is one of the mechanisms to counter SBS/ intestinal failure. Aging, recurrent infections, etc. can lead to the re-appearance of symptoms. Therefore, we must mention it as de-adaptation rather than a recurrence of SBS.
- I am pleased that you have done a literature search in Ichushi. However, a literature search must be performed in International databases, e.g. PubMed, etc. Please do a comprehensive literature search in one of the scientific databases.
- Grammatical errors need to be corrected.
Section-wise assessment/comments:
- Abstract- well written. Please consider using the term "re-emphasizing the concept of intestinal failure........"
- Introduction- well written.
- Case presentation
Line no 78-82, page no 2- please clear the past history. it is confusing. please rewrite regarding the past surgeries in chronological order.
Line no 93-94, page no 2- which pathological specimens are being referred here? Was this patient explored surgically? How was she treated finally? Is she alive?
- Descriptive overview- please revise the table. there is a lot of information in the special note, length of the bowel resected, and tracking period. I am more interested in the type of medical treatment provided and the survival status. Please reduce the already present columns and add the columns advised.
- Conclusions- You can not propose the concept of intestinal failure. You are merely "re-emphasizing" it. The review is incomplete till you give us information about the management of these cases.
Author Response
Dear authors
First, I would like to congratulate you for highlighting a new concept of recurrence of short bowel syndrome. Also, linking it to aging is another new concept. However, I have various comments that must be addressed.
Major issues
- First, Intestinal failure is not a new concept. In fact, it is quite commonly used in children. Please avoid using the words 'introducing the concept of Intestinal failure'. Rather, you are re-emphasizing it.
Response:
Thank you for highlighting this. We agree with this comment. Based on your suggestion, we have revised the abstract (Page 1, Line 23) and lessons learnt sections (Page 12, Line 179) by replacing the word “introducing” with “re-emphasizing”.
Second, the recurrence of SBS is again a very strong statement. I think you should mention it as "de-adaptation". Adaptation is one of the mechanisms to counter SBS/ intestinal failure. Aging, recurrent infections, etc. can lead to the re-appearance of symptoms. Therefore, we must mention it as de-adaptation rather than a recurrence of SBS.
Response:
Thank you for highlighting this. We agree with this comment. Based on your suggestion, we have revised the expression "recurrence" to "de-adaptation" throughout the manuscript.
I am pleased that you have done a literature search in Ichushi. However, a literature search must be performed in International databases, e.g. PubMed, etc. Please do a comprehensive literature search in one of the scientific databases.
Response:
Thank you for highlighting this point. We agree with this comment. However, we have focused on Ichushi in this review, in order to focus on Japan's super-aging society.
Grammatical errors need to be corrected.
Response:
Thank you for highlighting this. We agree with this comment. Based on your suggestion, we have obtained the services of a professional English language editing service, and thoroughly revised the manuscript for grammar and language.
Section-wise assessment/comments:
- Abstract- well written. Please consider using the term "re-emphasizing the concept of intestinal failure........".
Response:
Thank you for highlighting this. We agree with this comment. Based on your suggestion, we revised the sentence to, "Therefore, we stress the importance of re-emphasizing the concept of 'intestinal failure' in everyday practice, as well as other organ-related conditions such as cardiac or renal failure, as this may lead to a better understanding of the pathogenesis of malnutrition and diarrhoea in elderly patients." (Page 1, Lines 22-25)
- Introduction- well written.
Response:
Thank you for your comment, we appreciate it.
- Case presentation
Line no 78-82, page no 2- please clear the past history. it is confusing. please rewrite regarding the past surgeries in chronological order.
Response:
Thank you for highlighting this. We agree with this comment. Based on your suggestion, we have revised section 2. Case presentation; 2.1. Patient. We have rewritten the past history of the patient.
Line no 93-94, page no 2- which pathological specimens are being referred here? Was this patient explored surgically? How was she treated finally? Is she alive?
Response:
Thank you for highlighting this. We agree with this comment. Based on your suggestion, we have revised section 2. Case presentation; 2.1. Patient. The following sentences have been added:
“A colonoscopy was performed and pathological specimens were taken.” (Page 3, Line 109)
“Since diarrhoea and electrolyte abnormalities continued, we concluded that her bowel could not absorb nutrients, and therefore switched to central venous (CV) nutrition (delivered via a CV port), which improved her diarrhoea. She was discharged in good general condition about one month after creation of the CV port.” (Pages 3-4, Lines 113-116).
.
- Descriptive overview- please revise the table. there is a lot of information in the special note, length of the bowel resected, and tracking period. I am more interested in the type of medical treatment provided and the survival status. Please reduce the already present columns and add the columns advised.
Response:
Thank you for highlighting this. We agree with this comment. Based on your suggestion, we have revised Table 2. We have added the following new columns: “Type of medical treatment” and “Survival status".
- Conclusions- You cannot propose the concept of intestinal failure. You are merely "re-emphasizing" it. The review is incomplete till you give us information about the management of these cases.
Response:
Thank you for highlighting this. We agree with this comment. Based on your suggestion, we have revised the abstract (Page 1, line 24) and conclusion sections (Page 12, line 177) by changing the word, “introducing” to “re-emphasizing.”
Reviewer 2 Report
First of all thank you for this short overview about SBS, which is indeed a still lesser known field of gastro-enterology.
I have some minor remarks/questions I would like to see addressed:
1. Paragraph 3: Descriptive review:
As stated in the limitations of this study, the authors did a very narrow search on the subject. They limited this review to the research available in only one journal. Is there a specific reason why this search was so narrow, moreover since the subject is rather specific and rare. A broader search might have resulted in more information for the review.
2. The used literature is all in rather young patients, none older than 83 years. None of which aludes on the relationship between SBS and age. Hence, can you draw conclusions based on this used information? Can you please comment?
3. Paragraph 4: Discussion
The authors state that the lack of case reports is, in their opinion, based on the fact that the concept of SBS has only recently been taken care of. However, could the narrow search not be a factor in the limited number of case reports?
4.
Paragraph 4: Discussion
The authors state that the most probable factor responsible for this recurrence are the age-related changes in the intestines. On what grounds do you base this conclusions, since histological examination, coloscopy and CT did not reveal specific age-related pathology. I think this conclusion needs to be softened since correlation has not been tested for nor proven.
Author Response
First of all thank you for this short overview about SBS, which is indeed a still lesser known field of gastro-enterology.
I have some minor remarks/questions I would like to see addressed:
- Paragraph 3: Descriptive review:
As stated in the limitations of this study, the authors did a very narrow search on the subject. They limited this review to the research available in only one journal. Is there a specific reason why this search was so narrow, moreover since the subject is rather specific and rare. A broader search might have resulted in more information for the review.
Thank you for highlighting this. The reason for focusing on Japan was the possibility of the existence of many cases of intestinal failure in an aging society.
- The used literature is all in rather young patients, none older than 83 years. None of which aludes on the relationship between SBS and age. Hence, can you draw conclusions based on this used information? Can you please comment?
Thank you for highlighting this. I believe that the absence of elderly patients in this review means that intestinal failures may have been overlooked in an aging society.
- Paragraph 4: Discussion
The authors state that the lack of case reports is, in their opinion, based on the fact that the concept of SBS has only recently been taken care of. However, could the narrow search not be a factor in the limited number of case reports?
Thank you for highlighting this. We believe that the narrow scope of the search, as you mentioned, is a limitation of this study. We think it is necessary to increase the number of cases in the future for further study.
Paragraph 4: Discussion
The authors state that the most probable factor responsible for this recurrence are the age-related changes in the intestines. On what grounds do you base this conclusions, since histological examination, coloscopy and CT did not reveal specific age-related pathology. I think this conclusion needs to be softened since correlation has not been tested for nor proven.
Thank you for highlighting this. We agree with this comment. Based on your suggestion, we have revised the Discussion section by changing the statement “The most probable factor in the change from a stable state to an unstable state is the effect of ageing." to "SBS recurred, and the patient went from a stable to an unstable state. The cause of this change is not yet clear. However, we believe that it is due to the effects of aging. " (Page 11, Lines 154-155)
Reviewer 3 Report
I have difficulty understanding the theoretical point of view of the authors since, on the one hand they believe it necessary to introduce the concept of intestinal failure in the clinic as a real disease to be taken into consideration when intestinal symptoms (in particular diarrhea) occur but, on the other hand, they overlap it with SDS and aging.
I ask myself, and ask the authors for clarification, the following:
1) should intestinal failure be considered a clinical status regardless of age? Or not? 2) Should intestinal failure be considered a clinical status independent of SDS? Or not?
A clarification of these points would improve the whole manuscript.
Abstract
It is not clear the relationship the authors want to make between: a) aging and SBS; b) aging and intestinal failure; c) aging and intestinal function decline. They need to clarify these points.
Otherwise, the reader could only guess what the authors meant and ultimately the interpretation is wrong.
1.Introduction
The first paragraph should be better organized to improve reading avoiding repetitions.
All the chapter is conditioned by the ambiguities in the presentation of the different clinical situations. The last sentence, for example, from row 65 to row 68 make all even more difficult to understand since the authors write:
“..we report a case of recurrence SBS due to aging..” what do they mean?
“…provide a review of he current literature regarding recurrence of SBS.” And these cases go throughout infancy to aging
“… we discuss the relationship between intestinal failure and aging.”
Finally I was very confused what the authors want to demonstrate.
2.Case presentation
They do not need to add a subchapter as the present only one case.
Row 73: dii the patient underwent to intestinal resection 5 or 20 (as written in the abstract) years before the present hospitalization?
Row 93. Why they call the colonic specimens pathological? Di they look also to normal specimens?
4.Discussion
Row 130. Correct as follow “…she was symptom-free.” Not “they were”
Rows 133-134. The sentence starting with “that is, as an individual ages,…….” Is not clear.
Lessons learned
- Consideration for diarrhoea symptoms in the elderly
In this chapter there are several repetitions of what they already wrote in the introduction. Please synthesize and better organize this chapter
Author Response
I have difficulty understanding the theoretical point of view of the authors since, on the one hand they believe it necessary to introduce the concept of intestinal failure in the clinic as a real disease to be taken into consideration when intestinal symptoms (in particular diarrhea) occur but, on the other hand, they overlap it with SDS and aging.
I ask myself, and ask the authors for clarification, the following: 
1) should intestinal failure be considered a clinical status regardless of age? Or not? 2) Should intestinal failure be considered a clinical status independent of SDS? Or not?
A clarification of these points would improve the whole manuscript.
Abstract 
It is not clear the relationship the authors want to make between: a) aging and SBS; b) aging and intestinal failure; c) aging and intestinal function decline. They need to clarify these points.
Otherwise, the reader could only guess what the authors meant and ultimately the interpretation is wrong.
Thank you for highlighting this. We agree with this comment. Based on your suggestion, we have revised the Abstract section by including the statement, “ She had developed SBS once due to multiple surgeries, but due to compensatory function, the symptoms had abated. However, due to decreased intestinal function caused by aging, her ‘intestinal failure’ including SBS worsened, and symptoms reappeared.” (Page 1, Lines 17-20)
1.Introduction
The first paragraph should be better organized to improve reading avoiding repetitions.
All the chapter is conditioned by the ambiguities in the presentation of the different clinical situations. The last sentence, for example, from row 65 to row 68 make all even more difficult to understand since the authors write:
“..we report a case of recurrence SBS due to aging..” what do they mean?
Thank you for highlighting this. We agree with this comment. Based on your suggestion, we have revised the Introduction section by changing the term “We report a case of recurrence SBS due to aging.” to “We report a case of de-adaptation of SBS thought to be due to ageing,” (Page 2, Line 72).
“…provide a review of he current literature regarding recurrence of SBS.” And these cases go throughout infancy to aging
Thank you for highlighting this. We agree with this comment. Based on your suggestion, we have revised the Introduction section by including “To that end, for the first time in Japan, we report a case of de-adaptation of SBS thought to be due to ageing, as well as provide a review of the current literature on de-adaptation of SBS, to assess how many reports on de-adaptation of SBS with age have been published.”
“… we discuss the relationship between intestinal failure and aging.”
Thank you for highlighting this. We agree with this comment. Based on your suggestion, we have revised the Introduction section by changing “… we discuss the relationship between intestinal failure and aging.” to “Finally, we discuss the pathogenesis of the decline in intestinal function with aging, which leads to intestinal failure”
Finally I was very confused what the authors want to demonstrate.
2.Case presentation
They do not need to add a subchapter as the present only one case.
Thank you very much for highlighting this. We thought that a subsection was needed for the presentation of cases.
Row 73: dii the patient underwent to intestinal resection 5 or 20 (as written in the abstract) years before the present hospitalization?
Thank you for highlighting this. We have corrected the patient’s medical history based on your suggestion. Kindly also refer to Table 1 for details on the patient’s past history.
Row 93. Why they call the colonic specimens pathological? Di they look also to normal specimens?
Thank you for highlighting this. We agree with this comment. Based on your suggestion, we have revised the Case presentation section by changing the term “pathological specimen” to “colonic specimen.” (Page 3, Line 104)
4.Discussion
Row 130. Correct as follow “…she was symptom-free.” Not “they were”
Thank you for highlighting this. We agree with this comment. Based on your suggestion, we have revised the Discussion section by changing “they were” to “she was symptom-free.” (Page 11, Line 153).
Rows 133-134. The sentence starting with “that is, as an individual ages,…….” Is not clear.
Thank you for highlighting this. We agree with this comment. Based on your suggestion, we have revised the Discussion section by including, “The following are some of the mechanisms that may have contributed to the decline in bowel function with age, and thereby worsened intestinal failure.” (Page 11, Line 155).
Lessons learned
- Consideration for diarrhoea symptoms in the elderly
In this chapter there are several repetitions of what they already wrote in the introduction. Please synthesize and better organize this chapter
Thank you for highlighting this. We agree with this comment. Based on your suggestion, we have revised the Lessons learned section by including, “This change can lead to microbiota changes, increased intestinal permeability, and decreased absorptive function the accompanying overproduction of inflammatory cytokines such as IL-6, TNF-α, and IL-1β with ageing, thereby causing diarrhoea [7].” and “However, as mentioned above, diarrhoea may also be caused by disruption of intestinal bacterial flora and increased intestinal permeability due to persistent inflammation.”
Round 2
Reviewer 1 Report
I congratulate the authors for their work. All my comments have been addressed in the revised manuscript. The overall quality of the manuscript has significantly improved. The work has merit and will be of interest to our readers. Only a spell check is required for minor grammatical errors.